# Metabolic Dysfunction-Associated Steatotic Liver Disease Induced by Microplastics: An Endpoint in the Liver–Eye Axis

**DOI:** 10.3390/ijms26072837

**Published:** 2025-03-21

**Authors:** Ivan Šoša, Loredana Labinac, Manuela Perković

**Affiliations:** 1Department of Anatomy, Faculty of Medicine, University of Rijeka, 51000 Rijeka, Croatia; 2Department of Pathology and Cytology, General Hospital Pula, 52100 Pula, Croatia; loredana.labinac@obpula.hr (L.L.); perkovic.manuela@gmail.com (M.P.)

**Keywords:** cellular senescence, cytokines, environmental toxicology, eye, hepatokines, inflammation, liver–eye axis, toxicology

## Abstract

There is a significant, rather than just anecdotal, connection between the liver and the eyes. This connection is evident in noticeable cases such as jaundice, where the sclera has a yellow tint. But this can be seen through even more subtle indicators, such as molecules known as hepatokines. This relationship is not merely anecdotal; in some studies, it is referred to as the “liver–eye axis”. Ubiquitous environmental contaminants, such as microplastics (MPs), can enter the bloodstream and human body through the conjunctival sac, nasolacrimal duct, and upper respiratory tract mucosa. Once absorbed, these substances can accumulate in various organs and cause harm. Toxic substances from the surface of the eye can lead to local oxidative damage by inducing apoptosis in corneal and conjunctival cells, and irregularly shaped microparticles can exacerbate this effect. Even other toxicants from the ocular surface may be absorbed into the bloodstream and distributed throughout the body. Environmental toxicology presents a challenge because many pollutants can enter the body through the same ocular route as that used by certain medications. Previous research has indicated that the accumulation of MPs may play a major role in the development of chronic liver disease in humans. It is crucial to investigate whether the buildup of MPs in the liver is a potential cause of fibrosis, or simply a consequence of conditions such as cirrhosis and portal hypertension.

## 1. Microplastics and the Liver

Writing a review based on several articles related to the liver–eye axis and its endpoints was quite daunting and was made even more challenging when the topic of microplastics was included. To tackle this, we conducted a systematic review of the literature using three pairs of keywords in “ANY FIELD”: (1) “liver–eye axis” AND “microplastics”, (2) “steatohepatitis” AND “microplastics”, and (3) “(MAFLD” OR “MASLD”) AND “microplastics”.

To ensure comprehensiveness, we searched three databases via the Internet (PubMed, Web of Science, and Scopus), without any time constraints, from their inception to 6 March 2025. Most of the content accessed was peer-reviewed, i.e., evaluated by another expert in that subject area, so the information should have been of high quality. To prevent the accessing of misleading information that healthcare providers, policymakers, and other decision-makers may face due to an overwhelming volume of research, a systematic review of the scholarly literature was conducted.

Our initial screening identified 421 articles, with 63 removed as duplicates or non-English texts, which narrowed our search down to 358 entries. The study identification process was conducted on three levels: (1) inputs or the identification of studies, (2) processing or screening, and (3) outputs or included studies. This process is showcased in the flowchart in Figure 1.

### 1.1. Miceoplastics and Nanoplastics

Human development has led to numerous achievements, yet these advancements have often had a significant negative impact: environmental pollution. Critical trajectories of this process, such as food, air, water, and soil, have consistently affected human health. Additionally, the growth of human development has been closely linked to the creation of plastic waste, at least over the past few decades. The production of plastic waste more than doubled between 2000 and 2019, and this trend is continuing, particularly due to the substantial accumulation of plastics in rivers [1,2]. A large portion of this waste originates from plastics with a lifespan of less than five years, with 40% coming specifically from packaging [1].

Regardless of the duration, when plastic waste breaks down, it forms tiny particles known as microplastics (MPs) and nanoplastics (NPs), which are being increasingly found in human and animal tissues. Microplastics and NPs increase cytokine expression and induce enzymatic activity related to inflammation. Exposure to MPs can recruit neutrophils, macrophages, and natural killer cells to the liver [2].

In addition to being formed by the grinding of larger pieces of plastic waste, secondary MPs can also be produced in the form of primary MPs such as resin pellets used in plastic manufacturing. Ordinary consumer products, such as bisphenol A, which was once popular for use in toddlers’ bottles until a consumer boycott occurred [3], or chemicals like phthalates, are the primary sources of microplastic pollution in the environment [4,5]. In some literature, about 80% of the MPs detected were fibrous in shape and were made of polyethylene (25%), polyester (20%), and polyamide (10%) [6]. Microplastics (and NPs) originate from synthetic textiles, urban dust, tires, road markings, marine coatings, personal care products, and engineered plastic pellets [7,8]. Fine particles found in polluted air, such as micro- and nanoplastics, can cause systemic toxicity by penetrating cell membranes [9,10]. There is copious evidence that particle pollutants have caused a number of diseases or influenced existing ones (Table 1). Currently, plastic particles and fibers smaller than 1 μm are classified as nanoplastics, while those ranging from 1 μm to 1 mm are classified as MPs. Fragments that measure between 1 and 5 mm can be referred to as large MPs [11]. Based on all the reviewed literature, MP and NP particles are extremely complex and diverse in terms of their size, shape, density, polymer type, surface properties, and more.

Plastic particle pollutants are reported as risk factors for cardiovascular diseases and stroke. They are implicated in the etiopathogenesis of developmental disruptions, autism, and attention deficit hyperactivity disorder (ADHD) [12,13,14,15]. Plastics, particularly pollutants from the environment, precipitate a series of mental disorders. Additionally, plastic debris from environmental pollution is associated with cognitive decline in Alzheimer’s disease [16].

### 1.2. Liver Damage

Microplastic and NP particles activate the innate immune system, causing chronic, low-grade inflammation, which also enhances atherogenesis [17]. The NF-κB pathway is then activated [18,19], which enhances the inflammatory response in the liver [20]. Li et al., utilizing qRT-PCR, Western blotting, and immunohistochemical staining, confirmed that NPs/MPs induce liver fibrosis through alpha-smooth muscle actin (α-SMA) [21]. The protein expression level of α-SMA (not of mRNA) significantly increased as the size of the particles increased. It is reasonable to conclude that prolonged exposure to polystyrene microparticles and nanoparticles with larger sizes can significantly induce liver fibrosis.

Many studies have shown that the physicochemical properties of MPs can trigger inflammatory responses and provoke immune reactions when they enter the body and accumulate [22,23]. Once MP and NP particles enter the body, the host’s cells attempt to digest them using their lysosomes, and this ability is not limited to specialized phagocytes [24]. The host’s cells are usually unsuccessful, so plastic particles accumulate in their cytoplasm. Accordingly, after entering the liver tissue, these foreign bodies causes chronic, low-grade inflammation. The nuclear factor–kappa B (NF-κB) pathway is activated, further amplifying the inflammatory response in the liver [20]. When combined with signs of hepatocellular injury, such as the ballooning of hepatocytes, the condition is referred to as metabolic dysfunction-associated steatohepatitis (MASH) [25,26,27].

Regardless of whether this specification is characterized by fibrosis, it is referred to as metabolic dysfunction-associated steatotic liver disease (MASLD). MASLD is often asymptomatic and is usually diagnosed incidentally through abnormal liver enzyme findings or imaging studies that reveal hepatic steatosis. This is the initial stage of steatotic liver disease, which is characterized by more than 5 percent hepatic steatosis and at least one risk factor for cardiometabolic dysfunction, such as dyslipidemia or obesity [28,29,30]. Patients with MASLD should have minimal or no alcohol consumption and no other causes of steatotic liver disease [31]. As MASLD progresses, fat accumulation also continues and results in inflammation that damages the tissue (inflammation progresses from acute to chronic hepatitis), and scarring occurs (fibrosis) [32]. Fibrosis is a gradual process that advances through stages. In the end, it leads to cirrhosis: extensive scarring that disrupts liver architecture and function, significantly raising the risk of hepatocellular carcinoma (HCC).

Notably, the most widely supported theory on the pathogenesis of MASLD implicates insulin resistance coupled with systemic inflammation [33]. The excess accumulation of non-esterified fatty acids and diacylglycerol causes cellular distress and dysfunction, which leads to the overactivation of lipid signaling. This promotes hepatic insulin resistance and endoplasmic reticulum stress, resulting in chronic inflammation, liver fibrosis, cirrhosis, and, ultimately, hepatocarcinogenesis [34].

#### Chronic Inflammation, Oxidative Stress, and Cellular Senescence

Chronic inflammation, characterized by pro-inflammatory cytokines such as tumor necrosis factor (TNF) and interleukin 6 (IL-6), combined with oxidative stress, leads to liver damage [35,36]. The presence of polystyrene MPs significantly increases levels of TNF (formerly known as TNF-α), IL-6, and monocyte chemoattractant protein-1 (MCP-1), triggering oxidative stress and inflammation [37,38]. This suggests that a similar mechanism may occur in the liver [39,40]. Increased levels of IL-6 and TNF lead to inflammation and vascular permeability. Additionally, C-reactive protein (CRP), an acute-phase protein produced by the liver, has been linked to the development of both cardiovascular and ocular diseases, thereby connecting systemic inflammation to hepatic damage [36,41].

The pro-inflammatory nature of MPs is likely linked to cellular senescence [42], which is characterized by a permanent state of cell cycle arrest accompanied by the secretion of pro-inflammatory factors and other substances [43]. Various cell types exhibit increased senescence rates after exposure to different types of MPs and nanoplastics [44]. Similarly to oxidative stress and inflammasome activation, cellular senescence is increasingly recognized as a therapeutic target for drug development. Strategies to address this issue can promote the removal of senescent cells by either preventing their formation or inhibiting their harmful pro-inflammatory activity. For instance, Dhakal et al. [45] reported that a class of glucose-lowering medications can reduce senescence induced by MP and NP particles [45]. Notably, this same hypoglycemic medication has also demonstrated cardioprotective effects in various contexts. Another critical factor in cellular senescence is the overexpression of miR-217. This molecule has been shown to regulate critical processes, such as the cell cycle, differentiation, proliferation, and senescence [46]. miR-217 also plays a role in maintaining the structure and function of the extracellular matrix and might be relevant to the integrity of the liver [47]. Additionally, numerous studies have identified other effects associated with MP and NP particles, such as the induction of platelet aggregation, hemolysis, and the activation of immune cells [48].

## 2. Laboratory Assessment

### 2.1. Histology and Histologic Scoring Systems for Chronic Liver Disease

Liver biopsy remains the gold standard for confirming the diagnosis of MASH and assessing disease severity, and for accurately staging fibrosis [49]. Hepatic steatosis in MASLD—MASH—is characterized by hepatocyte ballooning degeneration and hepatic inflammation [50,51,52]. These histological features may be difficult to distinguish from those seen in alcohol-associated steatohepatitis [53]. Histologically, alcohol-associated steatohepatitis (ASH) is quite similar to MASH, making it challenging to distinguish between the two based solely on a biopsy. In MASLD, (the disease-specific histological criterion), hepatic steatosis features visible hepatocyte ballooning degeneration and hepatic inflammation [52].

Currently, non-invasive methods (non-invasive tests, NITs) for evaluating MASH are being studied. Circulating biomarkers are used in the NITs, and composite scores have been developed to combine the results of various NITs with other clinical parameters to predict the progression from steatohepatitis to fibrosis. However, their sensitivity or accuracy have not yet been proven, at least not enough to replace biopsy [54,55,56]. Only a liver biopsy can confirm a diagnosis of MASH, distinguish it from alcohol-associated steatohepatitis, assess disease severity, and accurately stage fibrosis [49].

In the arm of invasive tests and histologic sampling, non-alcohol-associated fatty liver activity score (NAS) is designed to assess the full range of changes associated with non-alcoholic fatty liver disease (NAFLD) and can be utilized to evaluate the effects of therapy. The total NAS can range from 0 to 8, with fibrosis not included in the score [57,58]. However, it is important to note that the threshold values for NAS do not always align with a diagnosis of MASH and should not be relied upon solely for diagnostic purposes [26]. The semiquantitative histological scoring system, also known as the Alcoholic Hepatitis Histologic Score (AHHS), is associated with the prognosis of patients with alcoholic hepatitis. This system utilizes four histological features: the degree of fibrosis, the degree of neutrophil infiltration, the type of bilirubin stasis, and the presence of megamitochondria. Additional histologic scoring systems for chronic liver disease include fatty liver inhibition and progression, and alcohol-associated liver disease and alcoholic hepatitis.

### 2.2. Assessing Microplastics

Tissue samples obtained by biopsy should be collected in glass tubes and frozen in liquid nitrogen, fixed in 10% buffered formalin, or fixed in 2.5% glutaraldehyde suitable for electronic microscopy for subsequent analyses.

#### 2.2.1. Macroscopy and Microscopy: Visual Assessment of Microplastics

Microplastics consist of various plastic particles with diverse compositions and sizes. [59]. The visual analysis method involves observing the samples, allowing the MPs to be roughly described and classified according to their color, shape, and size. Interest in quantifying MPs in environmental samples has grown, especially after stool and blood samples have documented exposure to MPs [60,61]. Proper analysis of the quality of MPs is most often performed via analysis using the naked eye or a microscope [62]. Although it cannot provide information on the chemical component of MPs [63,64], this analysis method is simple, inexpensive, and presents a negligible chemical hazard [65].

Data collection related to micro- and nanoplastics can be accomplished using various analytical methods. These include assessing particle number and size, determining polymer and additive identities, measuring mass fraction, and evaluating degradation status. However, no single method provides a complete picture, so a combination of different techniques with their respective limitations and advantages is necessary [64]. This is more challenging for nanoparticles because, without complete chemical information, it is unreliable to determine whether the quantified nanoparticles are nanoplastics at all [66,67]. Studies included in this review that assessed MPs visually identified them as having jagged edges and irregular shapes [68,69,70,71,72,73].

#### 2.2.2. Identifying Microplastic and Nanoplastic Particles

Pyrolysis–gas chromatography–mass spectrometry (Py–GCMS) is a widely favored method for identifying microplastic and nanoplastic particles, primarily due to its ability to detect non-volatile components in solutions, which may be overlooked in conventional GC-MS analyses. Additionally, Py–GC/MS can analyze a broader variety of sample types. This technique is also capable of detecting very small quantities of contaminants, down to microliters or micrograms within liters of sample material [74].

However, this method does not discriminate between microplastics and nanoplastics [75]. Furthermore, Py–GCMS cannot detect certain inorganic substances, and the results may vary slightly with non-homogeneous samples. Additionally, it cannot provide information on the number of particles across different size ranges. For information on particle sizes, Raman micro-spectroscopy or Fourier Transform Infrared (FTIR) spectroscopy are recommended. This variability may sometimes require repeated tests to obtain reliable results [75].

Our review found that studies predominately utilized Py–GCMS, with 83.3% (five out of six) of the included studies employing this method [69,70,71,72,73]. No studies utilized Raman spectroscopy or FTIR. Notably, Raman micro-spectroscopy is a technique that combines a Raman spectrometer, which analyzes the vibrational modes of molecules, with a confocal microscope. This method allows for the chemical analysis of samples with sub-micron spatial resolution, making it particularly effective in identifying MPs. In addition to sample identification, Raman micro-spectroscopy is also employed for quality control, failure analysis, material characterization, and examining physical and chemical properties [76]. FTIR analysis, or FTIR spectroscopy, is an effective method of assessing microplastics by identifying organic, polymeric, and, in some cases, inorganic materials. This method utilizes infrared light to scan test samples and observe their chemical properties. However, it was rarely discussed in the studies included in this review. Therefore, it remains speculative as to whether factors such as costs, staff limitations, the design of the sampling chamber, or the mounting of samples have contributed to the underutilization of FTIR spectroscopy [59,77].

One of the included studies used laser direct infrared imaging (LDIR) [71], which is an innovative infrared microscopy technique that employs a tunable Quantum Cascade Laser (QCL) as its infrared source. This new reflectance-based approach overcomes the coherence artifacts commonly associated with QCLs. It enables the acquisition of large-area, high-definition infrared images and high signal-to-noise point spectra. By extending this technique with attenuated total reflectance (ATR), it is possible to obtain high-fidelity spectra from features smaller than 10 μm.

In summary, Py–GC/MS can be utilized to analyze tissue samples specimens obtained from carotid or coronary arteries and samples from aortas.

### 2.3. Microplastic-Induced Histopathological Lesions in the Liver

Microplastics can alter the histological structure of tissues, including the liver. Feng et al. [78] conducted histological examinations of the livers of fish subjected to a diet containing varying concentrations of MP particles. They discovered that the fishes’ body weight was significantly reduced, and the antioxidant status of the liver was disrupted. However, in the group receiving the highest concentration of MP particles, only slight disorganization of the liver architecture was observed. The lack of morphological changes in the livers of the fish after MP exposure, which were significantly more pronounced in the MP-fed group than in the control group, indicates that factors other than the direct accumulation of MPs caused liver injury in the fish. More encouraging were the findings of Tian et al. [79], who observed several cellular changes in the H&E staining of liver tissue in zebrafish. They reported a loss of the cells’ polygonal shape, indistinct cell borders, vacuolar degeneration, and the infiltration of inflammatory cells. These lesions were attributed to the cumulative toxicity of MPs in the bloodstream and their subsequent impact on other cellular structures. At last, Zhuang and colleagues [80] identified liver inflammation, cell vacuolation, and karyopyknosis in the adult mice.

Li et al. observed liver injury in mice following exposure to nanoparticles and MPs [21]. They concluded that particle size significantly affected the impact of NPs and MPs on the liver. Notably, long-term exposure significantly decreased both liver weight and the liver-to-body weight ratio as the particle size of NPs and MPs increased. The observations of Li et al. also included hepatocellular vacuolar degeneration and edema, irregularly arranged hepatic cords, cell dikaryon, and inflammation. Similarly, Chen and colleagues found that non-uniform and irregular spherical particles lead to the accumulation of microplastics in duck livers [81].

The effects of microplastic exposure on the human liver were studied using a functionally active 3D liver microtissue model [82]. This model comprised primary human hepatocytes, Kupffer cells, sinusoidal endothelial cells, and hepatic stellate cells. Hematoxylin and eosin (H&E) staining revealed that the typical characteristics of the liver were preserved, with no necrotic core observed over the 21-day exposure period to MPs in the cell culture. This finding indicates adequate access to nutrients and oxygen in the inner zones of the microtissue, highlighting the suitability of this in vitro model for long-term toxicological studies.

Repeated low-dose exposure to smaller MPs led to distorted tissue architecture and the appearance of large extracellular voids, suggesting the presence of dilated canaliculi, a condition occasionally observed in the cholestatic liver. Additionally, intracellular white vacuoles were noted, resembling large lipid droplets, indicative of macrovesicular steatosis. Resident macrophages also accumulated non-biodegradable plastics, represented as uniformly sized intracellular white disks.

Horvatits et al. conducted a proof-of-concept case series that focused on detecting MP particles through the chemical digestion of tissue samples, staining with Nile red, and subsequent examination using fluorescent microscopy and Raman spectroscopy [83]. As a result, classical histopathological assessment was not performed. Their study primarily focused on the MPs, assessing their morphology, size, and the composition of the polymer types, while details regarding liver histoarchitecture were not provided. Nevertheless, the authors identified six different types of MP polymers in the livers of individuals with liver cirrhosis.

## 3. Toxic Exposure

Ocular toxicology focuses on the toxic effects of drugs administered through topical, intraocular, or systemic routes. It also includes the evaluation of adverse effects caused by ophthalmic devices, such as contact lenses, intraocular lenses, and glaucoma implants. With the emerging concept of the “liver–eye axis” and an increasing number of hepatokines influencing this relationship, the role of the liver in ocular toxicology is becoming increasingly significant [84,85].

Particulate matter from environmental pollution and other chemical substances (Figure 2), in the form of liquids, aerosols, or vapors (including medications), can breach the eye’s surface and its protective tear film [86]. In a local context, ocular surface damage can be assessed through a variety of ophthalmological symptoms that present in multiple ways. Toxic substances that affect the ocular surface can impair cellular motility and lead to cellular senescence [87,88]. An acute toxicity test used to evaluate the effects of chemicals and their potential to cause eye irritancy or damage is known as the Draize rabbit eye test. In this test, the effects of tested substances are observed using a slit-lamp biomicroscope on enucleated rabbit eyes. The test is not only controversial from the perspective of animal rights but also raises concerns about the subjective scoring of effects and the differences between rabbit and human eyes. For this reason, its scientific validity has been questioned [89]. To be precise, the test observes and records (using a subjective scoring system) effects on the conjunctiva, cornea, and iris. These effects can vary from mild and reversible irritation to severe and irreversible damage that may result in vision loss [90]. To address these shortcomings, a test based on a 3D corneal tissue model was developed and assessed in a series of validation studies. It is currently the subject of OECD test guideline No. 491 [91,92].

These particular pollutants from the air are associated with causes of systemic inflammatory response and cytokine production [93]. Millen et al. [94] performed a scoping review of the scholarly literature to determine whether ambient air pollution was a risk factor for chronic disease of the inner ocular structures and elevated intraocular pressure. They examined 27 identified articles where air pollutants were considered responsible for eye conditions. In their scoping review, Millen et al. [94] examined scholarly literature to assess whether ambient air pollution was a risk factor for chronic diseases affecting the inner ocular structures and elevated intraocular pressure. Their review identified 27 articles that linked air pollutants to various eye conditions. The systemic effects of airborne pollutants that enter the body through the ocular surface are well documented and do not require further discussion here. However, the limited literature on this topic indicates a significant need for additional research.

Environmental toxins lead to oxidative damage and inflammation. Air pollution can reduce the thickness of the tear-film lipid layer, as demonstrated in a study by Wang et al. [95] involving 200 healthy volunteers over a period of three years. This reduction can impair the motility of corneal and conjunctival cells, potentially leading to their senescence or even apoptosis. Consequently, this disruption affects important tissue barriers that regulate ocular drug absorption [96].

Fine particles, such as MPs and NPs, can lead to systemic toxicity by penetrating cell membranes. There is substantial evidence that these particle pollutants are responsible for a variety of diseases or exacerbate existing health conditions. Plastic particle pollutants have been identified as risk factors for cardiovascular diseases and strokes. Additionally, they are associated with developmental disorders, including autism and attention deficit hyperactivity disorder (ADHD) [12,13,14]. Plastic waste, including alternatives to traditional non-degradable plastic polymers, is linked to causes of systemic inflammatory responses and cytokine production [97,98]. Plastic and other environmental pollutants can lead to various mental disorders. Additionally, plastic debris from environmental pollution is linked to cognitive decline in Alzheimer’s disease [16].

Ongoing tests of environmental pollutants and their harmful effects utilize organoids and organoid-on-chip technologies, which leverage their advantages while addressing the limitations of traditional approaches. Although this research is still in its early stages, it demonstrates excellent potential for advancing environmental toxicology [99].

In summary, toxic substances from the eye’s surface can lead to local oxidative damage by inducing apoptosis in corneal and conjunctival cells [100,101]. Irregularly shaped microparticles can exacerbate this effect. Additionally, toxicants from the ocular surface may be absorbed and distributed through the bloodstream. Environmental toxicology presents a challenge because many pollutants can enter the body via the same ocular surface route as certain medications [102].

## 4. Liver–Eye Axis

There is no irrefutable evidence that MPs or NPs enter the bloodstream over the conjunctiva–environment interface, but this review offers a non-negligible number of reports of environmental pollutants damaging deeper eye structures after being in contact with that interface [97,102,103]. Also, there is a plethora of evidence of the ubiquity of MPs and NP particles [4,104], similar to the growing popularity of fluids from the OS as a biomonitoring tool or a prospect toxicological matrix [85,105]. The recent identification of MPs in the olfactory pathway suggests a potential major entry route for plastic into the brain; thus, the OS should not be dismissed lightly [106].

In that context, the liver–eye axis highlights the connection between these organs as a new area of research that examines the interconnected pathways through which the liver and eyes affect each other’s functions in both health and disease. In recent decades, the number of biomolecules secreted by the liver that regulate systemic metabolism, known as hepatokines, has steadily increased. The effects of these molecules on distant organs often include the eyes [107,108,109]. Hepatokines in the liver–eye axis include fibroblast growth factor 21 (FGF-21), hepatocyte growth factor (HGF), adropin, and angiopoietin-like proteins (ANGPTLs), which have emerged as key factors [110,111,112,113] (Table 2). As inflammation is the underlying cause of MASLD, inflammatory mediators such as CRP, IL-6, and TNF may hold significant importance in the liver–eye axis [33,114]. In the experiment conducted by Zhuang et al. [80], exposure to microplastics in adult mice resulted in increased levels of TNFα and IL-1β, as measured by qRT-PCR.

The connection between the liver and the eyes has recently gained more attention, although it primarily focuses on a one-way relationship. The concept of the “liver–eye axis” is not new nor merely a trend; it is deeply rooted in the theories and practices of Traditional Chinese Medicine (TCM). This perspective could provide a scientific basis for researchers interested in studying the liver–eye connection in a holistic way [135,136]. While this area deserves further investigation, it is also essential to consider the potential liver toxicity associated with ethnomedicine [113,137]. This concept should also be promoted in Western clinical practice. In this way, the latest advances in understanding the liver–eye axis could also be applied to toxicology. This connection has gained recognition even beyond TCM. It can be achieved through various pathways, including inflammation, immunity, metabolism, and oxidative stress [138,139].

The communication that relies on secretory factors and associated cytokines primarily involves hepatokines, molecules secreted exclusively or predominantly by the liver. These molecules are mainly delivered to the liver through paracrine activity or can reach distant organs via the human circulatory system. Additionally, some factors secreted by the eyes can also influence the liver’s condition, acting at a distance. For instance, the receptors of hepatocyte growth factor (HGF) have been detected in the cornea, lens, and retinal tissues [140]. On the other hand, retinal pigment epithelium-derived factor (RPEDF) excreted by the retinal pigment epithelium (RPE) may influence the liver [141]. There have been an increasing number of epidemiological studies indicating that eye disease is associated with protective or risk factors related to liver health. The prompt identification of these factors could improve healthcare outcomes. This review used this information as proof of concept.

## 5. Conclusions

In summary, toxicants from the eye surface that cause oxidative damage to the ocular surface may be absorbed into the bloodstream and distributed throughout the body. This should be noted when discussing the etiopathogenesis of MASLD and when administering MASLD medications. Consequently, all MASLD medications should be tested in clinical trials that examine the potential liver–eye connection, including any possible negative effects of these drugs on eye health. Improving our understanding of this interconnectedness may ultimately enhance patient outcomes by enabling more precise preventive and therapeutic measures to address the underlying metabolic causes affecting both liver and ocular health.

Future studies are needed to evaluate whether the hepatic accumulation of MP from the ocular surface is a potential cause of fibrosis or a consequence of cirrhosis and portal hypertension. In diagnosis and toxicological assessment, artificial intelligence and deep-learning-based diagnostic tools for detecting conditions related to the liver–eye axis could be a future direction for analytical toxicology.

## Figures and Tables

**Figure 1 ijms-26-02837-f001:**
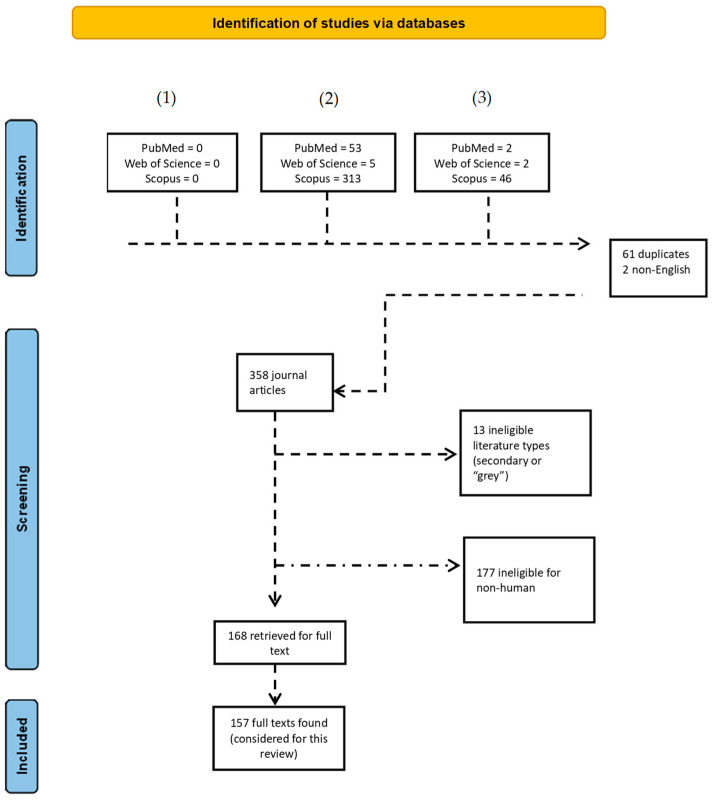
This diagram illustrates the study selection process and is based on the Preferred Reporting Items for Systematic Reviews and Meta-Analyses (PRISMA) statement. It is important to note that the International Prospective Register of Systematic Reviews (PROSPERO) does not accept scoping reviews, literature reviews, or mapping reviews. Therefore, this literature review was not registered with PROSPERO.

**Figure 2 ijms-26-02837-f002:**
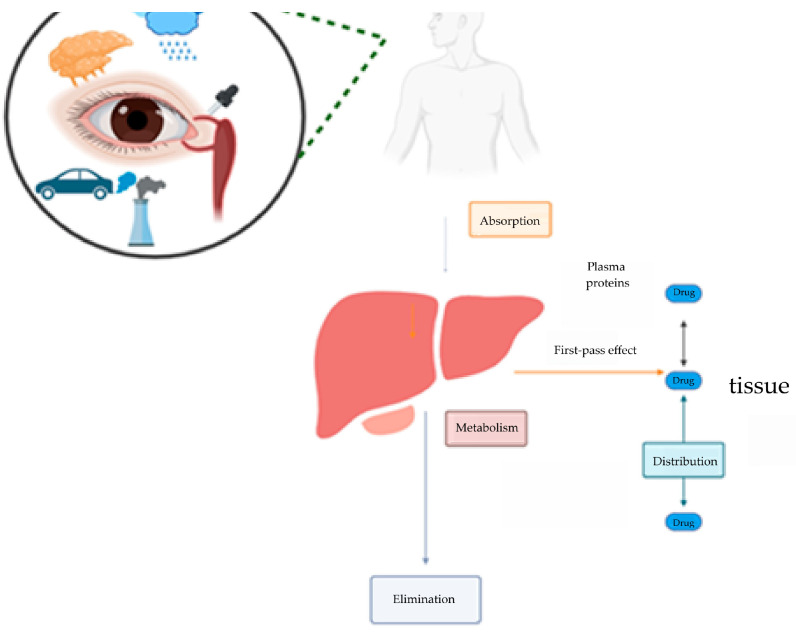
Schematic of substances entering the system through the eye and then being distributed after the first-pass effect. (BioRender licensed to the University of Rijeka, Croatia, https://www.biorender.com (accessed on 29 January 2025), was used to make the figure).

**Table 1 ijms-26-02837-t001:** The classification of plastic waste depending on size.

Plastic Debris Name	Particle Size
**Macroplastic** **s**	>5 mm
**Larger microplastics**	1–5 mm
**Microplastics**	67–500 µm, 1–5000 µm, 20–5000 µm, or more broadly, as <5000 µm [definition supported by the National Oceanic and Atmospheric Administration (NOAA)]
**Nanoplastics**	<20 µm to as small as 1 nm

**Table 2 ijms-26-02837-t002:** Hepatokines with the annotation of those involved in the liver–eye axis.

Hepatokines	Correlated with (Biological Functions)	Liver Is the Target Organ/Paracrine Mechanism?	Impacts the Liver–Eye Axis? Possible Key Factors for Liver–Eye Contact	References
Adropin	Macronutrient intake and estrogen.	Yes	Yes	[112,113,115]
ANGPTL3	Increases lipid production in the liver and plasma lipid levels, promotes lipogenesis and the liver’s inflammatory response, and decreases glucose uptake.	Yes	Yes	[112,113,116,117]
ANGPTL4	Inhibits lipoprotein lipase and activates cAMP-stimulated lipolysis in adipocytes.		Yes	[112,113,118]
ANGPTL6	Enhances insulin signaling in skeletal muscle and mitochondrial oxygen consumption in white adipose tissue. Inhibits gluconeogenesis in the liver.	Yes	Yes	[112,113,119]
ANGPTL8/betatrophin	Dubious action on beta-cell proliferation.	Yes	Yes	[112,113,120]
CFH	A complement inhibitor inhibits its excessive activation and is a key player in maintaining complement homeostasis. It is present as a soluble protein and is also attached to cell surfaces throughout the human body.	Yes	Yes	[110,111,121,122]
Fetuin-A	Increases inflammation and insulin resistance.	Yes		[123,124,125]
Fetuin-B	Increases hepatic steatosis and mediates impaired insulin action and glucose intolerance.	Yes		[125,126]
FGF-21	An insulin-sensitizing hormone/metabolic actions.	Yes	Yes	[112,113,127]
Follistatin	Increases insulin resistance, promotes thermogenesis, and induces the differentiation of brown adipocytes. Enhances glucose levels and the uptake of free fatty acids after exercise training. Inhibits FSH production and suppresses skeletal muscle growth.	Yes		[128,129]
GDF15	Increases energy metabolism and lowers body weight; it is possibly implicated in the pathogenesis of anorexia. Stimulates thermogenic and lipolytic genes. Improves glucose tolerance and insulin sensitivity. Prevents liver steatosis.	Yes		[110,111,113,122]
Hepassocin	Promotes insulin resistance and adipogenesis.	Yes		[110,111,122]
HGF	Paracrine cellular growth, motility, and morphogenic factor.	Yes	Yes	[110,111,112,113,122,130,131]
LECT2	Promotes the accumulation of lipids and inflammation in the liver and the development of insulin resistance in skeletal muscle.	Yes		[110,111,122,132]
RBP4	Depending on the source, the effect may be controversial. It increases lipolysis in adipocytes, is associated with insulin resistance and components of metabolic syndrome, promotes hepatic mitochondrial dysfunction and hepatic steatosis, and impairs insulin signaling.	?	?	[110,111,112,122,133]
Selenoprotein P	Inhibits hepatic gluconeogenesis and decreases glucose uptake in the skeletal muscle.	Yes		[122]
SMOC1	Improves glycemic control via inhibiting gluconeogenesis and glucose output from the liver.	Yes		[110,111,122,134]

ANGPTLs: antiopoietin-like proteins; CFH: complement factor H; FSH: follicle-stimulating hormone; GDF15: growth differentiation factor 15; HGF: hepatocyte growth factor; LECT2: leukocyte cell-derived chemotaxin 2; RBP4: retinol-binding protein 4; SMOC1: SPARC-related modular calcium-binding protein 1.

## Data Availability

The data used in this paper are available upon request.

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
