# Peer review of "Metabolic Dysfunction-Associated Steatotic Liver Disease Induced by Microplastics: An Endpoint in the Liver–Eye Axis"

_ijms, 2025, doi:10.3390/ijms26072837_

Round 1
Reviewer 1 Report
Comments and Suggestions for Authors
General Comments:
This review explores the developing and highly relevant area of microplastics (MPs) as environmental pollutants and their possible role in the pathogenesis of Metabolic Dysfunction-Associated Steatosis Liver Disease (MASLD). The authors also introduce the idea of the "liver-eye axis," highlighting the organized pathways between the liver and eyes in health and disease. The review is well-structured, complete, and offers a detailed review of the current works, including toxicological mechanisms, histological findings, and the role of hepato-kines in the liver-eye axis. However, there are areas where the review could be improved to scientific rigor, enhance clarity, and overall impact.
Major Comments:
1. Novelty and Scope:
o The manuscript provides a novel perspective by linking microplastic exposure to MASLD and the liver-eye axis. However, the linking between microplastics and the liver-eye axis stays underexplored. While the authors discuss the liver-eye axis in detail, the straight evidence connecting microplastics to this axis is limited. Establishment this linking with more attentive discussion or suggesting specific methods would enhance the review's impact.
2. Mechanistic Insights:
o The review discusses the role of inflammatory pathways (e.g., cytokines like TNF-α, NF-κB, and IL-6) in MASLD pathogenesis. Though, the mechanistic pathways linking microplastics to these inflammatory responses could be explained further. For example, how do microplastics exactly activate these pathways in the hepatic organ, and how might this change from other environmental toxins?
3. Histological Findings:
o The part of histological findings is thorough but could benefit from a stronger comparison between microplastic-induced hepatic damage and other causes of MASLD (e.g., metabolic or alcohol-related). Moreover, the discussion of histological marking systems (e.g., NAS, FLIP) is detailed but rather tangential to the main attention of the review. Consider shrinking this section to maintain attention on microplastic-induced changes.
4. Liver-Eye Axis:
o The liver-eye axis is an intriguing concept, but the manuscript does not provide sufficient evidence to establish a strong connection between microplastics, liver dysfunction, and ocular health. While the discussion of hepatokines is valuable, the manuscript would benefit from more specific examples or experimental data linking microplastic exposure to ocular outcomes via the liver-eye axis.
5. Toxicological Assessment:
o The manuscript highlights the challenges of assessing microplastic toxicity and mentions advanced models like organoids and organoid-on-chip technologies. Expanding on how these models could be used to study the liver-eye axis or microplastic-induced MASLD would strengthen the discussion.
Minor Comments:
1. Abstract:
o The abstract is strong but could be briefer. Consider focusing on the key findings and implications rather than providing extensive related information.
2. Tables and Figures:
o The manuscript references a schematic figure (Figure 1) illustrating the distribution of substances entering the system through the eye. However, the figure is not included in the provided context. Ensure that all figures are included and clearly labeled.
3. Terminology:
o The manuscript uses multiple terms for liver diseases (e.g., MASLD, MASH, MetALD). While these terms are defined, their frequent use can be confusing. Consider streamlining the terminology for clarity.
4. References:
o The manuscript is well-referenced, but some citations (e.g., [87-90]) are clustered together without clear differentiation. Ensure that each reference is appropriately cited and linked to specific statements.
5. Language and Grammar:
o The manuscript is generally well-written but contains minor grammatical errors and awkward phrasing (e.g., "Be it as it may"). A thorough proofreading is recommended.
Recommendations for Improvement:
1. Strengthen the Connection Between Microplastics and the Liver-Eye Axis:
o Provide more direct evidence or propose specific mechanisms linking microplastic exposure to the liver-eye axis. This could include a discussion of how microplastics might influence hepatokines or ocular health.
2. Focus on Microplastic-Induced MASLD:
o While the manuscript provides a comprehensive overview of MASLD, the focus on microplastic-induced MASLD could be sharpened. Consider condensing sections on general MASLD pathogenesis and histological scoring systems.
3. Propose Future Directions:
o The manuscript briefly mentions the potential use of AI and organoid models in toxicological assessment. Expanding on these future directions would add value and demonstrate the authors' forward-thinking approach.
4. Clarify the Role of Hepatokines:
o The discussion of hepato-kines is valued but could be more intensive. Highlight specific hepato-kines that are most relevant to the liver-eye axis and microplastic exposure.
General Assessment:
• Significance: High. The review reports an important and emerging topic with potential effects for environmental toxicology, hepatic disease, and ocular health.
• Originality: Moderate. While the liver-eye axis is a novel concept, the connection to microplastics requires further development.
• Clarity: modest. The review is well-organized but could benefit from more brief writing and a sharper attention on the main topic.
• Scientific Rigor: Moderate. The manuscript provides a thorough review of the literature but lacks direct experimental evidence to support some of its claims.
Comments on the Quality of English Language
The manuscript is generally well-written but contains minor grammatical errors and awkward phrasing (e.g., "Be it as it may"). A thorough proofreading is recommended.
Author Response
Thank you for all your comments.
In terms of novelty and scope, the liver-eye axis is now more clearly connected to microplastics, providing solid evidence linking these two focuses. Additionally, this connection is backed by a more thorough discussion.
The pathways linking microplastics to inflammatory responses have been explained more clearly. The processes of cellular senescence and phagocytosis have been introduced to enhance understanding.
The manuscript is now organized around the theme of "laboratory assessment." This includes a histological evaluation of liver tissue, highlighting instances where microplastics are responsible for liver changes. This approach allows for better comparison between microplastic-induced hepatic damage and other causes of Metabolic Associated Steatotic Liver Disease (MASLD). Section 2 includes a subsection titled “2.2. Assessing Microplastics.”
Due to a lack of literature on the liver-eye axis, more specific examples or experimental data linking microplastic exposure to ocular outcomes via this axis are also limited.
The same authors have previously published work on the methodology of sample collection in ocular toxicology. To avoid self-citation, they chose not to include this information in the current submission.
The language and grammar have been reviewed by a professional service (certificate attached), ensuring that the manuscript is free of any minor grammatical errors identified by the reviewers.
Reviewer 2 Report
Comments and Suggestions for Authors
Dear Authors,
I read carefully your manuscript “Metabolic Dysfunction-Associated Steatotic Liver Disease (MASLD) Induced by Microplastics: An Endpoint in the Liver-Eye Axis”.
My first opinion was that it was courageous to write the review based on a couple of articles directly connected with the endpoint of the article's liver-eye axis. But, reading the manuscript it turned out that the first aim of linking MASLD induced by microplastics could be the basics for possible conclusion.
The manuscript is well-written and organized with some parts which don’t belong there.
The Abstract is too direct and not quite based on the facts reading in the manuscript. I could not find an article that directly indicates that microplastics can enter the human body through the conjunctival sac.
The segment numbered one bring into connection MASLD and microplastics but not through the liver-eye axis. The presence of polystyrene induced an increased level of cytokines in mice kidneys. Could you please explain how this mechanism could be adapted to the liver?
The part 1.1 is very interesting and indicates the connection between microplastics and liver, by recruiting neutrophils, macrophages and natural killer cells in the liver. Could you recognize the potential mechanism of microplastics entering the organism? The effects of microplastics on systemic health are very well explained. They concluded that prolonged exposure to microplastics can induce liver fibrosis.
Regarding histological findings, the authors tried to differentiate MASLD-MASH, alcoholic hepatitis, primary biliary cholangitis, and primary sclerosing cholangitis. They evaluated the scoring system, and they did it in detail.
Linking microplastics with histopathological lesions in the organism, showed that it alters the histological structure of tissues, including the liver. Could you please explain the direct connection regarding MASLD and microplastics? You definitely showed the interaction between liver and microplastics through changes in the liver of fish, mice and human 3D liver microtissue model.
Are you sure that between pollutants entering the eye and inducing severe ocular surface damage, are microplastic particles? Which particles did you observe? Are all of them connected to the systemic inflammatory response? Can MP and NP lead to systematic changes by penetrating the ocular membrane? Is there evidence for that, or we can still speculate about that possibility?
Could you please explain why specific hepatokines emerge as key factors for liver-eye contact? What kind of connection is it?
You showed that they are molecules secreted by eyes directly influencing the liver, and reversed, the receptor for HGF has been detected in the ocular tissue. Could you suppose that these substances positively or negatively influence the liver-eye axis?
I quite agree with the conclusion that future studies are needed to evaluate whether the hepatic accumulation of MP from the ocular surface is a potential cause of fibrosis or a consequence of cirrhosis and portal hypertension. Also, all MASLD medications should be tested in clinical studies in order to examine the potential liver-eye connection, including any possible negative effects of these drugs on eye health. I believe that you can conclude that I consider the conclusion as the best part of the manuscript.
Best regards
Author Response
Thank you for all your comments.
We revised the abstract to make it less direct and more focused on the facts presented in the manuscript. To alleviate concerns about the challenge of writing a review based on a limited number of articles directly related to the endpoint of the liver-eye axis, we adopted a strategic approach standard for systematic reviews when conducting our literature search. This strengthened the foundation for our conclusions. As a result of these changes, the introductory part of the manuscript has been completely rewritten and reorganized.
The section on histological findings has also been modified (we omitted unnecessary discussions about the scoring system, which had been detailed elsewhere) to focus on the identification of microplastics. The resulting subsection (2.2) now interposes a general assessment of MASLD with the areas where microplastics are identified. This approach informs the reader about the assessment of microplastics before linking them with histopathological lesions.
Specific hepatokines that emerge as key factors in the liver-eye connection are indicated in Table 2. Additionally, since the literature lacks an article that directly demonstrates that microplastics can enter the human body through the conjunctival sac, we reference a similar finding related to olfactory nerves and the brain.
Reviewer 3 Report
Comments and Suggestions for Authors
The manuscript by Perkovic et al is a review of how microplastics can induce MASLD. The first section defines MASLD and microplastics. This section really describes the effect of microplastics on inflammation pathways and some physiological consequences such as ADHD and AD. The second section is about histology of MASLD. The 3rd section is about toxic exposure mainly from the eye to liver. The fourth section is the establishment of the liver-eye axis and the induction of hepatokines involved with inflammation of liver and/or eyes. This is followed up with a long table of hepatokines along with references. Section 5 is the conclusion of the authors.
Some suggestions to improve the paper include:
Section 1: The description of MASLD is sufficient. The part on microplastics and nanoplastics should be revised and re-organized (1.1). What is the difference between a microplastic and nanoplastic? What are their compositions? What are common sources? How do they come in contact with people and animals? Where are they most prevalent in the environment? Polystyrene is mentioned, but are there other chemistries that are common? It may be worth looking at papers sponsored by the American Chemical Society in that they have a lot of information on micro/nanoplastics.
Section 2: A page and a half is devoted on scoring and diagnosis of MASH and not used else where in the paper. This may reduced to one paragraph and situated in section 1. Part 2.2 is good and more examples may be used with a focus on size, chemistry, and route of exposure.
Section 3: Some of the assertions that microplastics may cause cognitive disorders need to be more rigorously examined. Currently, it is difficult to treat the brain with pharmaceutical drugs as the blood brain barrier (BBB) keeps most of these away from the brain. How do micro/nano particles which are much bigger than molecules get into the brain? Another issue is that aerosol pollutants may get in the body via the eye. However, the surface area of the aveola of the lung is much much higher and the exposure to aerosol plastics would also be much higher.
Section 4: Liver health has effect on eye health. The author cite review articles #84 and 85. They need to cite the actual research papers. For example, FGF-21 in the table should have a reference number matching #86 in review article #85 in their reference list.
Section 5: The first sentence “toxicants from the eye surface cause oxidative damage to the ocular surface and may be absorbed to the bloodstream and distributed through body.” Doing a thorough literature review should yield a more definitive conclusion. Give a % risk assessment. It seems that liver disease induced partially by micro/nanoplastics may affect eye health, but from what I can perceive is that the amount of toxins coming in through the eye may be negligible and more pronounced if these particles are inhaled or ingested.
Author Response
Thank you for your comments.
To clarify the difference between microplastics and nanoplastics, we have added Table 1 along with a brief descriptive paragraph outlining their differences and similarities. This includes common sources of these particles and how they come into contact with both humans and animals.
Round 2
Reviewer 3 Report
Comments and Suggestions for Authors
The paper is very much improved in this revision. It flows better and is more focused on the theme of the review.